# Delivery, setting and outcomes of paediatric Outpatient Parenteral Antimicrobial Therapy (OPAT): a scoping review

Bernie Carter,[1] Enitan D Carrol,[2] David Porter,[7] Matthew Peak,[3] David Taylor-Robinson,[4] Debra Fisher-Smith,[2] Lucy Blake[1]

¹Faculty of Health and Social Care, Edge Hill University, Ormskirk, UK
²Department of Clinical Infection, Microbiology and Immunology, University of Liverpool, Liverpool, UK
³Paediatric Medicines Research Unit, Alder Hey Children's Hospital, Liverpool, UK
⁴Department of Public Health and Policy, University of Liverpool, Liverpool, UK
⁷Department of Infectious Diseases and Immunology, Alder Hey Children's Hospital, Liverpool, UK

**Correspondence to**
Professor Bernie Carter;
bernie.carter@edgehill.ac.uk

## ABSTRACT

**Background** There has been little detailed systematic consideration of the delivery, setting and outcomes of paediatric Outpatient Parenteral Antimicrobial Therapy (OPAT), although individual studies report that it is a safe and effective treatment.

**Objective** This scoping review aimed to examine what is known about the delivery, settings and outcomes of paediatric OPAT and to identify key knowledge deficits.

**Design** A scoping review using Arksey and O'Malley's framework was undertaken.

**Data sources** Keywords were identified and used to search MEDLINE and CINAHL.

**Study appraisal methods** Primary research studies were included if samples comprised children and young people 21 or under, who had received OPAT at home or in a day treatment centre. The Mixed Methods Appraisal Tool was used to review the methodological quality of the studies

**Main findings** From a preliminary pool of 157 articles, 51 papers were selected for full review. 19 studies fitted the inclusion criteria. Factors influencing delivery of OPAT were diverse and included child's condition, home environment, child-related factors, parental compliance, training and monitoring. There is little consensus as to what constitutes success of and adverse events in OPAT.

**Conclusions** Future studies need to clearly define and use success indicators and adverse events in order to provide evidence that paediatric OPAT is safe and effective.

**Implications** Consensus outcomes that include child and parent perspectives need to be developed to allow a clearer appreciation of a successful paediatric OPAT service.

## BACKGROUND

Children with serious bacterial infections (SBIs) have been treated using parenteral antimicrobial therapy in an outpatient setting since the mid-1970s.[1] At this time, the intramuscular route was considered to be a clinically safe and largely successful means of treating infection.[2] However, advances in intravenous therapy and the requirement to protect children from the pain associated with the intramuscular injections led this route to fall into disfavour.

**Strengths and limitations of this study**

► Identification of methodological weaknesses in studies.
► Identification of gap in knowledge about parents and children's experience of Outpatient Parenteral Antimicrobial Therapy (OPAT), the lack of predetermined success criteria and clarity about what constitutes an adverse event.
► Due to the variable quality of the evidence base, strong conclusions regarding the delivery, settings and outcomes of OPAT cannot be made.

More recently, paediatric Outpatient Parenteral Antimicrobial Therapy (OPAT) has been defined as the parenteral administration of antimicrobials for at least two consecutive days without an intervening hospitalisation.[3] This treatment is selectively offered to treat SBIs such as pneumonia,[4] osteoarticular infections[5] and low-risk febrile neutropenia.[6] Depending on the child's condition at presentation, the child may be admitted to hospital and receive initial treatment and monitoring until deemed sufficiently stable to be discharged home on OPAT, or the child may be referred immediately for OPAT without ever having been admitted to hospital.

Two main approaches to OPAT delivery are used depending on the local resources available. The first is ambulatory and requires the child to return to a clinical setting (eg, emergency department or day treatment centre) on a daily (or more frequent) basis for assessment and administration of the therapy.[7 8] The second approach is home based, with the child being assessed and the therapy administered in the child's home either by nursing staff or by their parents who have been trained to assess and administer the antibiotics.[4] When the service is delivered by nurses, this is usually undertaken by those who are

**Table 1** Inclusion and exclusion criteria

| Inclusion criteria | Exclusion criteria |
| --- | --- |
| 1. Primary research studies.<br>2. Articles in peer-reviewed journals.<br>3. Published in English.<br>4. Data are presented from children and young people aged 21 years or under (and are reported separately from adults' data).<br>5. Children and young people who received Outpatient Parenteral Antimicrobial Therapy treatment did so in their home or a day treatment centre and data from inpatients and outpatients were reported separately.<br>6. Children and young people received at least 80% of treatment intravenously.<br>7. Data from intramuscular and intravenous treatment reported separately. | 1. Studies conducted in developing/low-income settings.<br>2. The full text of the article was unavailable.<br>3. Case studies, reviews, guidelines, poster, abstracts, commentaries and editorials. |

either part of a specialised OPAT team of community nurses,[9 10] or those within a broader community role such as 'hospital at home'.[11 12]

A variety of patient and healthcare benefits are potentially associated with OPAT; most notably for health services is that OPAT is considered to be a more cost-effective option when compared with continued inpatient care.[13] Other benefits include 'parent and patient satisfaction, psychological well-being, return to school/employment, reductions in healthcare-associated infection and cost savings' (Patel *et al*, p361).[3]

Given that there has been little consideration of the direct and indirect benefits, disadvantages and broader outcomes of paediatric OPAT, a scoping review was conducted to examine what is known about OPAT in terms of delivery, settings and outcomes, and to identify key areas of deficits in knowledge. Specifically, this scoping review explored primary research that examined OPAT delivered to children and young people aged 21 or under, who had received OPAT in a home or day

treatment centre, of which at least 80% of treatment was intravenous.

## METHOD
A scoping review was undertaken as the intention was to explore and map the key concepts and to identify gaps in research related to paediatric OPAT. The scoping review was conducted following Arksey and O'Malley's framework,[14] which was modified to allow more flexible and robust reporting of the results.[15–17] These modifications included: (1) an iterative approach to refine our search strategy and inclusion criteria; (2) an assessment of methodological quality was undertaken using the Mixed Methods Appraisal Tool (MMAT)[18] and (3) in the absence of EQUATOR guidance on reporting we were guided by recommendations made by the Joanna Briggs Institute.[19]

### Inclusion criteria and types of sources
The inclusion and exclusion criteria are shown in table 1. No date restrictions were applied to the search.

The search terms were generated based on consideration of: the population (children and young people under the age of 21 years), the 'concept' under investigation (parenteral antimicrobial treatment) and the context (home-based or outpatient-based care). Keywords and terms identified by the authors were used to search PubMed and CINAHL. Further keywords were then identified and the new search list was used to search Google Scholar to generate a comprehensive final set of search terms (table 2).

### Search strategy
Major databases consulted for the indexed published literature were MEDLINE and CINAHL. Further articles not identified in the results of the above strategies were added if identified by other means (eg, cited by a related article, identified on a World Wide Web search). The search was initially undertaken in February 2017 and updated in July 2017 and was supported by an expert librarian (full electronic search strategy available on request). A data extraction sheet was developed and iteratively refined and included the following broad categories: delivery, setting and outcomes. In line with the aims of a scoping review, all outcomes of paediatric outpatient treatment were included in the data extraction sheet.

### Appraisal of study quality
The MMAT[18] was used to review, but not score, the methodological quality of the studies. In 7 of the 19 studies, it

**Table 2** Search terms (by population, concept, context)

| OPAT<br>OR paediatric outpatient parenteral antimicrobial therapy | Population (<21 years) | ► Paediatric OR pediatric OR infant OR child* OR adolesce*<br>► Infection OR infectious disease |
| --- | --- | --- |
| | Concept (Intervention) | ► Antibiotic OR antimicrobial AND (agent OR therapy OR prescri* OR manage*)<br>► Parenteral OR intravenous infusion OR home infusion |
| | Context (Setting) | ► Outpatient OR home OR ambulatory OR community |

was not completely clear that the collected data adequately allowed the research question to be answered. Other key quality issues related to completeness of outcome data, appropriateness of measurements and acceptability of response rate (see table 3).

## RESULTS
### Overview of the studies

A preliminary pool of 157 articles were identified. Titles and abstracts were reviewed by two lead reviewers and where there was disagreement a third reviewer was used; 51 papers were selected for full review, from which 19 were identified as having good fit with the inclusion criteria and the objective of the review (see figure 1).

A condensed summary chart detailing the study design, sample, requirements, setting and delivery of the 19 studies included in the review is available as an online supplementary file. The review of studies via the MMAT revealed the quality as fair (see table 3).

Data were international, reporting on studies undertaken in the USA,[5 8 10 13 20–23] Canada,[6 7 9] Spain,[11 24] Australia,[12 25] Ireland,[4] India,[26] Israel[27] and the Netherlands.[28]

The studies included in the review had adopted a record review design with the exception of three cohort studies,[7 11 12] two randomised control trials,[25 26] an online survey[20] and a pilot programme.[6]

All study populations comprised children and/or young people aged 1 week to 21 years, with the exception of one study which presented data from an online survey of paediatric physicians.[20] Sample sizes ranged from 7[28] to 2687.[21]

### Delivery of service: target population, indications for treatment, factors influencing delivery

The studies were mixed in terms of whether the studied cohort had a common underlying condition as well as the infective indication for receiving OPAT. Only four studies had a specific focus on one such condition: cancer[6 25] and cystic fibrosis.[4 28] The remaining studies had either no specific underlying condition reported[5 7–10 21–23] or the children had a range of underlying conditions (such as gastrointestinal diseases and HIV infection).[11–13 20 24 26 27]

In terms of the infective indications for treatment, half of services delivered OPAT for a wide range of infection (eg, respiratory, blood stream, urinary and musculo-skeletal).[8 10–13 20–23] The remaining half were focused on a single indication for treatment, such as urinary tract infections.[4–7 9 24–28]

The key consideration in determining the suitability of the child for OPAT was the presence of infection. Other factors included the stability of the child's condition[7 9 11 24 26] and the home environment, either in generic terms[20] or more specifically such as the need for the home environment to be 'stable'[10] and appropriately resourced in terms of refrigerator and/or telephone.[8 10 11 25 26] The location of the home was specifically reported as influential in determining access to OPAT by four studies[11 25–27]

and, although, this was not clearly reported, it is likely that this was relevant in other studies where specialist home-based teams delivered OPAT.

Parental compliance/reliability was also reported as either an inclusion factor[8 11 20 26] and/or the lack of these qualities as an exclusion factor.[9 10] Parents were trained to administer medication to their child in six studies.[4 6 11 22 24 27] In all of these studies, all children had a pre-existing condition. However, even when professionals were responsible for the administration of medication, parents received training to: assess for complications[4 10 11 22 28]; to check the child's temperature,[9 26] deterioration[27]; inspect the intravenous site[8] and troubleshoot.[6] Five studies reported that training parents required a period of time in hospital before discharge to OPAT[4 10 11 22 28] and one study reported that a period of hospitalisation was needed to check for drug reactions.[23]

Support for parents or carers varied across the studies depending on whether the child was receiving home-based or ambulatory care. For children in the home setting, support varied from daily phone calls and home visits as needed,[11] initial daily or twice daily visits,[12 22 25] visits about every 2.9 days[27] and 24 hours access to professional support.[4–6 27] For children receiving ambulatory-based OPAT, parents were advised to return to the emergency department and/or readmitted if they had concerns.[8 25]

### Setting

In most studies, the family home was the setting for the delivery of OPAT.[4 6 10–13 22–25 27 28] The remaining studies were set in various outpatient settings: day treatment centres,[7 9] a combination of hospital outpatient/local clinics[26] or emergency department.[8] In three studies, the location was not reported or unclear.[5 20 21] Little detail was provided about the outpatient settings or the actual suitability, difficulties or challenges of the home as a setting for OPAT.

### Outcomes

No studies reported a priori criteria for success of paediatric OPAT. 'Success' was therefore implied in terms of the percentage/number of children completing OPAT as home based or outpatients[13] or through reports of what percentage of episodes of treatment were completed at home.[11] Other studies claimed that home treatment improved the child's condition compared with previous hospital-based courses of treatment,[28] or implied success through noting that all ambulatory patients returned for scheduled re-evaluation within 24 hours of commencing OPAT/initial discharge.[8]

Clinical complications such as line failures, rehospitalisation and adverse drug reactions (ADRs) were not consistently reported as adverse events (AEs) although these have been identified as such in the Summary Chart.

The reporting of hospitalisation/readmission was inconsistent. Although some studies reported the number of children who were hospitalised after commencing

**Table 3** MMAT synthesis table

| Type of study | Methodological quality criteria | Randomised Controlled Trial | | Quantitative descriptive | | | | | | | | | | | | | | | | |
|---|---|---|---|---|---|---|---|---|---|---|---|---|---|---|---|---|---|---|---|---|
| | | Gupta et al[26] | Orme et al[25] | Banerjee et al[20] | Campo et al[24] | Peláez Cantero et al[11] | Doré-Bergeron et al[9] | Gauthier et al[7] | Glackin et al[4] | Goldman et al[21] | Gomez et al[22] | Hodgson et al[12] | Le et al[23] | Madigan and Banerjee[10] | Maraqa et al[5] | Reid and Bonadio[8] | Shemesh et al[27] | Van der Laag and Van de Weg[28] | Van Winkle et al[13] | Wiernikowski et al[6] |
| Screening questions | Are there clear research question(s)? | Y | Y | Y | Y | Y | Y | Y | N | Y | Y | Y | Y | Y | Y | Y | Y | N | Y | Y |
| | Do the data address the research question? | Y | Y | Y | Y | Y | Y | UC | UC | Y | UC | UC | Y | Y | Y | Y | Y | UC | UC | UC |
| RCT | Is there a clear description of randomisation? | Y | Y | | | | | | | | | | | | | | | | | |
| | Is there a clear description of concealment? | N | N | | | | | | | | | | | | | | | | | |
| | Are there complete outcome data? | UC | Y | | | | | | | | | | | | | | | | | |
| | Is there low withdrawal? | Y | Y | | | | | | | | | | | | | | | | | |
| Quant. Descript. | Is the sampling strategy relevant? | | | Y | Y | Y | Y | Y | Y | Y | Y | Y | Y | Y | Y | Y | Y | Y | Y | Y |
| | Is the sample representative? | | | Y | Y | Y | Y | Y | Y | Y | Y | Y | Y | Y | Y | Y | Y | N | Y | UC |
| | Are measurements appropriate? | | | UC | Y | Y | Y | Y | UC | Y | UC | Y | Y | Y | Y | UC | Y | UC | UC | UC |
| | Is there an acceptable response rate? | | | N | Y | Y | Y | Y | UC | UC | UC | Y | Y | Y | Y | Y | Y | UC | Y | UC |

MMAT, Mixed Methods Appraisal Tool; N, no; Y, yes; UC, unclear.

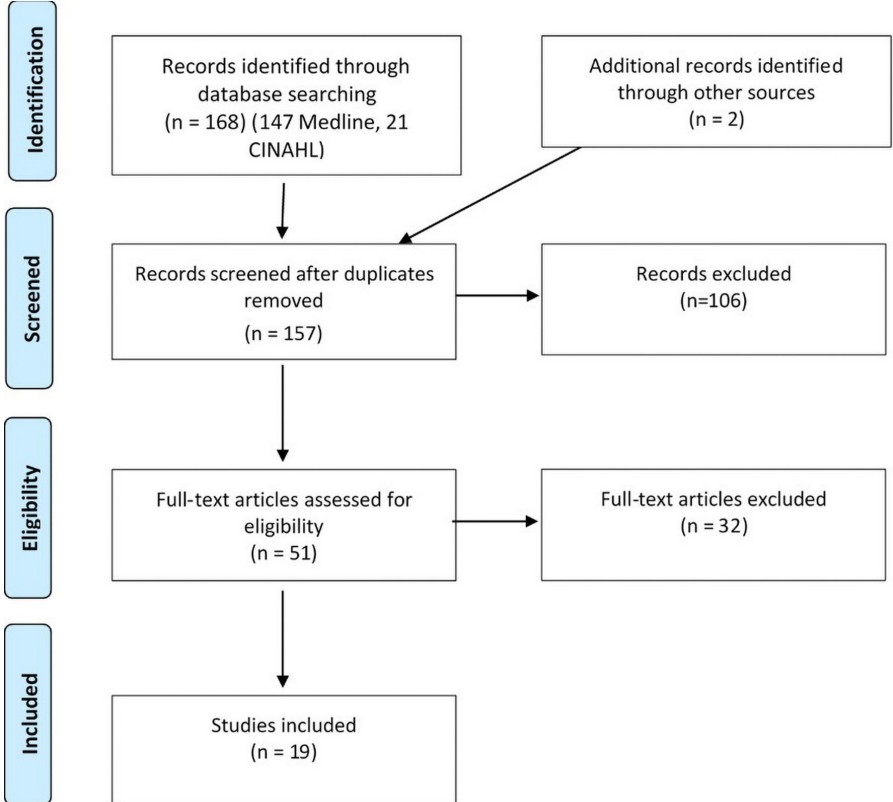

**Figure 1** Flow diagram for scoping review process (from Joanna Briggs Institute manual).

OPAT,[3 7 9–13 22 23 25 26] others reported the number of treatment courses that required unplanned hospitalisation.[5 21 24] Hospitalisation rates varied ranging from 4%[12] to 22%[27] of patients and between 26%[5] and 29%[22] of courses. Children were hospitalised as a result of being 'unresponsive to treatment' (22%)[27], 'inadequate clinical response' (1%)[12], 'exacerbation of underlying condition' (7%)[11], 'poor evolution of infectious disease' (3%),[11] 'deterioration' (0.6%),[4] fever[24 26] and the need to 'complete course of intravenous'.[22] Catheter associated complications were also linked to hospitalisation.[5 13 21 23 24] Other reasons for hospitalisation included ADRs and surgical management,[5 21] seizures and bleeding,[26] gastro-oesophageal reflux and positive blood culture result.[9] In two studies the reason for admission was less clear.[7 10] Unplanned medical care visits were reported in two studies with 17 out of 98 (17%) children having an unplanned visit[23] and 17 (48%) having one or more unplanned visits.[10]

The number of other catheter-related complications was reported by five studies[5 12 22 23 27]; only one study reported no catheter-related complications.[6] Extravasation, displacement and other intravenous access issues were reported by 5 studies.[7–9 11 28] Poor technique and/or technical problems were reported by two studies.[11 28]

The definitions and reporting of ADRs was inconsistent between studies. Four studies reported that no ADRs occurred,[6 8 25 27] and others provided generic reports of ADRs. For example, 25% of children experiencing OPAT complications were reported as being associated with the use of highly bioavailable antibiotics.[21] Two studies

(11%) provided more detailed reports of ADRs: in one study, ADRs were associated with 70 (29%) courses and, of these, early discontinuation of antibiotics was reported in 58 courses of treatment[22] and in the other study ADRs were associated with inappropriate choice of drug (6%) and inappropriate dose or duration of treatment (26%), although the authors also reported that no adverse antibiotic-related events necessitated change or cessation of antibiotic or hospital readmission.[12]

Seven studies reported on satisfaction (parental satisfaction[6 7 25 27 28], children's satisfaction[27 28]), although the mechanisms of data collection were often unclear or unreported. In one study, some parents (32%) were worried about taking their febrile child home and 20% were worried about taking their child home with indwelling intravenous access.[7] In another study, some mothers of children aged 6–12 years were anxious about accepting the responsibility of their child's treatment and concerned about the stress that home-based care would create for the family.[28] The 12–18 year olds in this study described liking home-based care due to the lack of disruption to home and school life, but reported missing the contact with staff and other patients that occurred when they were inpatients. In another study, children aged 10 or over completed questionnaires assessing their quality of life. Those who were treated at home had significantly better appetites and slept better compared with those who were treated in hospital.[25]

Six studies concluded that OPAT is more cost-effective than conventional inpatient treatment[6 7 10–13]; two studies

noted that the cost-effectiveness calculations did not account for the costs associated with complications[10] or the direct cost(s) to families.[6]

## DISCUSSION

This scoping review has systematically examined the empirical evidence regarding the delivery, settings and outcomes of paediatric OPAT. The quality review revealed that the studies are generally fair quality. The operationalisation of specific definitions/treatments varied widely and the reporting of who gave treatment and the setting was often unclear.

The factors influencing the delivery of OPAT were diverse and included: service-related issues including staffing and monitoring; child-related factors such as age, nature of infection, clinical status; and home/parent-related factors such home environment, parental compliance and training.

In a systematic review comparing home-based versus hospital-based treatment with intravenous antibiotics in children, the authors concluded that data about the safety of treatment were scarce.[29] In addition to a scarcity of data, this review found that there is a lack of clarity and consensus as to what constitutes success in OPAT making comparison across studies difficult; however, individual studies report that OPAT is safe.[5–7] [12 27] There is also a lack of clarity and consensus in the definition and reporting of AEs. There was little acknowledgement that although problematic, defining AEs is necessary or acknowledging that for one type of AE -ADRs- objective criteria do exist and could be used. Conclusions about the success of OPAT have been drawn despite evidence of AEs (which were ill defined, yet occurred in most studies) and readmissions (which were reported in different ways, and likewise occurred in most studies).

In terms of key knowledge deficits within the literature we scoped for this review, most of the studies were retrospective and follow-up data examining health outcomes over time are lacking. We also know little about parents and children's experience of OPAT. There is little reflection about the factors which may influence experience such as the child's age, nature of infection, family circumstances and the educational level of parents. Additionally, considering the fact that infection has a higher incidence in families of lower socioeconomic status,[30 31] there is little detail about whether these families are excluded from OPAT or, if in receipt, how they fare in comparison to families in better circumstances. We know little of children who were not selected for OPAT or parents and children who declined this treatment and the reasons why. In agreement with the recent systematic review comparing home-based versus hospital-based treatment, we likewise conclude that although studies report patients to be safely treated at home, generalisation to all patients is difficult due to selection bias[29]

The evidence base for the economic benefits of OPAT is poorly and inconsistently presented and does not take account of any shift of economic burden onto the families.

### Strengths and limitations

This scoping review has used a robust and iterative methodological approach and included an analysis of study quality. However, the variable quality of the evidence base means that strong conclusions regarding the delivery, settings and outcomes of OPAT for children cannot be made. Conclusions are also complicated to draw due to the diversity in terms of the age of children receiving treatment, children's underlying conditions, indications for treatment and the delivery of treatment. Our focus was outpatient care, therefore, our findings do not reflect comparison with inpatient care.

### Implications for research

Future studies need to clearly define success indicators and AEs in order to substantiate claims that OPAT for children is safe and effective. Specifically, hospitalisation, unexpected catheter-related complications, extravasation and antibiotic complications should be reported as AEs. To allow comparison between studies and pooling of data from different cohorts, the definitions for such AEs need to be agreed by healthcare professionals delivering adult and paediatric OPAT care.

Numerous knowledge deficits need to be addressed. There is a need for follow-up data tracking the trajectory of patient's interactions with healthcare providers over time. Future research of a qualitative nature needs to be conducted with children and young people receiving OPAT, and their parents in order to explore their experiences of receiving this treatment. A thorough cost-benefit analysis needs to conducted that includes a consideration of the economic impact on the family.

### Implications for practice

Parental and child perspectives should be sought to identify how they can best be supported. Despite the apparent professional confidence in the success and benefits of OPAT for children, it should not be assumed that all families will choose OPAT or that it will be the most appropriate intervention. Clear, consensus outcomes that include outcomes of importance to the children and their parents need to be developed to allow a clearer appreciation of a successful OPAT service.

## CONCLUSION

Further work that includes the perspectives of children and parents and which uses clearly defined indicators will improve the evidence base for the efficacy and safety of pOPAT.

## PATIENT AND PUBLIC INVOLVEMENT

Neither patients nor the public were involved in the scoping review process.

**Contributors** All authors (BC, EC, DP, MP, DT-R, DF-S and LB) contributed to the conception and design of the study. Article reviewing, scoring and data analysis has been performed by BC and LB with assistance from EC. All authors (BC, EC, DP, MP, DT-R, DF- S and LB) have made contributions to the drafting and revision of the article.

**Funding** LB, DP and DF-S were part funded by The National Institute for Health Research Collaboration for Leadership in Applied Health Research and Care North West Coast (NIHR CLAHRC NWC) for undertaking this review. BC received no funding. DTR is funded by the MRC on a Clinician Scientist Fellowship (MR/P008577/1).

**Disclaimer** The views expressed are those of the author(s) and not necessarily those of the NHS.

**Competing interests** None declared.

**Patient consent** Not required.

**Provenance and peer review** Not commissioned; externally peer reviewed.

**Data sharing statement** The summary chart has been provided as an online supplementary table.

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
