## [Reviewer comments · BMJ Open]

ARTICLE DETAILS

TITLE (PROVISIONAL)	THE DELIVERY, SETTING AND OUTCOMES OF PAEDIATRIC OUTPATIENT PARENTERAL ANTIMICROBIAL THERAPY (pOPAT): A SCOPING REVIEW
AUTHORS	Carter, Bernie; Carrol, Enitan; Porter, David; Peak, Matthew; Taylor-Robinson, David; Smith, Debra; Blake, Lucy

VERSION 1 – REVIEW

REVIEWER	Asha Bowen Princess Margaret Hospital for Children, Perth, Australia
REVIEW RETURNED	05-Feb-2018

GENERAL COMMENTS	This is a well written paper using a different methodology to reach similar conclusions to the pOPAT Guidelines published by Patel et al in JAC in 2015 and also Bryant et al in Lancet Infectious Diseases in 2017. The point of difference is the methodology used and I think that this is a useful addition to the literature albeit it very similar to the systematic review in Lancet iD. It provides an overview of the programmatic approach to pOPAT from the published literature and in doing so questions the published results about measurement of outcomes, adverse events and costs. It would benefit from an additional paragraph on the design and methodology for future research in pOPAT in order to bridge this gap. Please address referencing line 14, page 3 (ref 5 looks to be incorrect as it refers to overseas adoption and not pOPAT), line 55, page 3 (ref not inserted), line 45 page 7 (ref not inserted), line 20, page 8 (reference not inserted to feature in reference list)
---

REVIEWER	Penelope Bryant The Royal Children's Hospital Melbourne, Australia
REVIEW RETURNED	02-Mar-2018

GENERAL COMMENTS	This is a review of studies of OPAT in children investigating delivery, setting and outcomes. As OPAT in children increases, reviews and syntheses of data from primary studies are useful. It appears to have an appropriate design and the writing style is clear and easy to follow. Although the authors state that this is the first review to systematically examine outcomes of paediatric OPAT, there was a systematic review published last year on this topic (outcomes, although not delivery and setting). However, more importantly, the criticisms by the authors of this review of the studies they have
---

included are often unwarranted as they criticise lack of data that the original studies did not aim to provide. The statements are often true, but not based on the data that has been reviewed. There also appear to be some missing studies, although the methodology is not completely clear about this. This has resulted in a review that is lopsided and occasionally inaccurate, and in some places the tone is even condescending which is a real pity as reviews on this topic are needed.

Major points

‘However, there has been no detailed systematic consideration of the direct and indirect benefits, disadvantages and broader outcomes of pOPAT.’

‘This is the first study to systematically examine the empirical evidence regarding the delivery, settings and outcomes of pOPAT. Unfortunately it isn’t. See Bryant & Katz Lancet Infectious Diseases 2017 ‘Inpatient versus outpatient parenteral antibiotic therapy at home for acute infections in children: a systematic review’

There are some unique factors assessed in this study such as delivery and setting, but these statements should be modified.

Why was inclusion criteria ‘received at least 80% of treatment intravenously’? ie why exclude a patient with osteomyelitis who received 2-3 weeks of OPAT followed by 3-4 weeks oral antibiotics?

The methodology is unclear regarding:

- 1) Whether the included studies had to have all three components of information on delivery, setting and outcomes to be included
- 2) Which outcomes they were interested in a priori (the authors are critical of the included studies for not stating their definitions of outcomes in the methods)
- 3) Did the 33 studies that were not ‘a good fit’ all fulfil the exclusion criteria in table 1? There are some studies in the Bryant & Katz systematic review that seem as though they should be included in this review from the inclusion criteria, but this may not be the case – it is just not completely clear from the methods.

Because the methods are not clear on these points, it is therefore unclear whether the methodology is appropriate to answer the aim, especially the aim about key deficits in knowledge. ie any deficits found may have been published in other studies that were not included in this review’s search.

There appears to be at least one missing study that should have been included by Orme et al Outpatient versus inpatient IV antibiotic management for pediatric oncology patients with low risk febrile neutropenia: a randomised trial. *Pediatr Blood Cancer*. 2014 Aug;61(8):1427-33. The primary and clear outcome was parent and patient experience with the sample size specifically calculated for that, and this may modify some of the discussion in this review.

The authors state that ‘Clinical complications such as line failures, rehospitalisation and adverse drug reactions were not reported as adverse events although these have been identified as such in the outcomes data extraction table’, which appears to suggest that OPAT services are not reporting adverse events. This is inaccurate

eg in Hodgson et al 'Table 5 Adverse events' specifically reports line complications, antibiotic complications and unplanned readmissions to hospital. It is not clear whether this is an oversight from one included study, whether this has occurred for multiple studies, or whether the authors mean something different.

Readmissions to hospital are among the most critical outcomes when assessing an OPAT service, but the paragraph on this seems incomplete for several reasons:

- the authors only categorize unresponsive to treatment, exacerbation of underlying condition and poor evolution of infectious disease, but there are other reasons for readmission reported in several studies eg line complications, antibiotic complications, an unrelated reason – why are these not included?
- could the authors explain the difference between unresponsive to treatment and poor evolution of infectious disease
- the above categories are only reported for 3 studies but these are reported in other studies too – why are the other studies not included?

The paragraph on cost analysis is also flawed for a couple of reasons:

- 'None of the studies undertook a comprehensive health economic evaluation' – since this is not the aim of a clinical description of an OPAT service, while true, this is an unwarranted criticism. A good health economic analysis would be an entirely different study. It's akin to saying none of the studies assessed variation in care using a health services delivery approach – true, but not the aim of the studies
- 'or were systematic in their approach' – could the authors clarify what this means and by whose definition? They may have been systematic but did not include all related costs eg costs to families
- 'although these conclusions should be considered with caution' – this may be true but needs an explanation why?
- 'However, cost effectiveness calculations did not account for the costs associated with complications (26) or the direct cost to families (6).' Is this true of all the studies that attempted to evaluate cost or just the two cited?

'There is a lack of clarity and consensus as to what constitutes success in pOPAT, although this has not deterred authors from concluding that pOPAT appears to be safe (6,8,10,23,24).'

This sentence is again unnecessarily critical and comes across as condescending. If a service has low adverse events, low OPAT-related complication rates, low readmission rates and high rates of completion of treatment then why would the authors of this review not respect the authors of the original study's conclusion that OPAT is safe, even if their definition of success is different from another service or study?

The paragraph on key knowledge deficits is true, but again, if these areas were not aimed to be addressed by the included studies, this is an unwarranted criticism:

- 'We also know little about parents and children's experience of pOPAT. There is little reflection about the factors which may influence experience such as the child's age, nature of infection,

family circumstances and the educational level of parents.’ – this wasn’t the aim of many if the included studies; in a study discussing clinical outcomes for cardiac patients, the parent and child experience would not be a requirement of a study unless it was a stated aim.

- Additionally, considering the fact that infection has a higher incidence in families of lower socio-economic status (28,29), there is little detail about

whether these families are excluded from pOPAT or, if in receipt, how they fare in comparison to families in better circumstances. – again, while this is important to know, this wasn’t the aim of the included studies; in a study on the efficacy of an antibiotic to treat a particular infection, a socio-economic analysis would not be required if this wasn’t the aim of the study, despite infections having a higher incidence in lower socio-economic groups.

Minor points

Is this called a scoping review because it isn’t a PRISMA-compliant systematic review? (although the authors have stated systematic a couple of times)

pOPAT is an unnecessary and non-universal abbreviation. Once the authors have identified that the study is in children, the term OPAT can be used.

Suggest having the condensed summary chart as a supplementary table rather than ‘available on request’.

Suggest adding the type of quantitative study (eg record review, cohort, etc) to table 3.

When listing the numbers of studies that reported particular factors eg home environment criteria, it would be more useful to the reader of the references to the specific articles could be included with statements so the reader can review these themselves. Without this, the reader is unable to relate the study with the specific factors addressed. Studies are cited in some sections but not others – it is inconsistent.

Two other limitations:

- 1) That the authors would have been unable to determine whether some factors may not have been identified in a study because this was either not included by the authors of the study or editor of the journal it was published in as too obvious to include (eg parental compliance or suitable home environment) or not considered to need inclusion for other reasons, rather than that the service not providing it.
- 2) That the authors did not address comparison with inpatient care – although this was not stated as an aim, it should be mentioned in this paragraph as it is a potential limitation of any OPAT study without it

‘Little detail was provided about the outpatient settings or the actual suitability, difficulties or challenges of the home as a setting for pOPAT’

	If this was not the aim of the included study this is an unwarranted criticism. 'Adverse drug reactions (ADRs) were reported inconsistently and the definition of what constituted an ADR was not clearly defined.' This is not accurate. In some studies what constituted an ADR was very clearly defined, but may have been different between studies or not clearly defined in others. Suggest change to 'The definitions and reporting of ADRs was inconsistent between studies.' 'We also know little of children who were not selected for pOPAT or parents and children who declined this treatment and the reasons why.' This is in the paragraph on study quality but is unrelated to quality: a study can have high quality but if this is not one of the aims of the study then this is an unwarranted criticism. 'it should not be assumed that all families will choose pOPAT or that it will be the most appropriate intervention.' Did any of the studies state this, or is the authors' own assumption? The conclusion 'Future studies need to clearly define and use success indicators and adverse events in order to provide evidence that pOPAT is safe and effective' is unnecessary as the almost identical sentence appears three paragraphs above. Neutropenia not neutropaenia. There is no reference for the RCT by Gupta et al. Inconsistent use of whole numbers or one decimal place for percentages.
--	--

VERSION 1 – AUTHOR RESPONSE

Submission due: 1st April (word count 4000 words)		
	We would like to thank the reviewers for their insightful comments and suggestions that have undoubtedly improved the clarity of this scoping review. Please see the table below for details as to how we concerned each issue that has raised.	
	Issue	
1	include a PRISMA checklist and state the page numbers where each item can be found. The checklist can be downloaded from here: http://www.prisma-statement.org/documents/PRISMA%202009%20checklist.pdf	A PRISMA checklist has been included with the submission.
2	- Please include the search dates for the study in the methods section. Was this from inception of the databases? To what date?	No date restrictions were applied to the search.
3	It provides an overview of the programmatic approach to pOPAT from the published literature and in doing so questions the published results about measurement of outcomes, adverse events and costs. It would benefit from an additional paragraph on the design and methodology for future research in pOPAT in order to bridge this gap.	The paragraphs in which future research is addressed on page 14 and 15 have been amended to be more specific regarding study design.

4	Please address referencing line 14, page 3 (ref 5 looks to be incorrect as it refers to overseas adoption and not pOPAT), line 55, page 3 (ref not inserted), line 45 page 7 (ref not inserted), line 20, page 8 (reference not inserted to feature in reference list)	The references have been checked and errors have been amended.
5	'However, there has been no detailed systematic consideration of the direct and indirect benefits, disadvantages and broader outcomes of pOPAT.' 'This is the first study to systematically examine the empirical evidence regarding the delivery, settings and outcomes of pOPAT. Unfortunately it isn't. See Bryant & Katz Lancet Infectious Diseases 2017 'Inpatient versus outpatient parenteral antibiotic therapy at home for acute infections in children: a systematic review' There are some unique factors assessed in this study such as delivery and setting, but these statements should be modified.	This review was published after our scoping review search was conducted. We've not included it as it is a review, rather than primary research, but we have referred to it in the paper.
6	Why was inclusion criteria 'received at least 80% of treatment intravenously'? ie why exclude a patient with osteomyelitis who received 2-3 weeks of OPAT followed by 3-4 weeks oral antibiotics?	In the absence of clear external benchmarks to use as an inclusion criterion, this was discussed amongst the team and based on clinical judgment. 80% was selected a reasonable cut-off point.
7	1) Whether the included studies had to have all three components of information on delivery, setting and outcomes to be included	No. The studies did not need to include all three elements.
8	2) Which outcomes they were interested in a priori (the authors are critical of the included studies for not stating their definitions of outcomes in the methods)	As this was a scoping review (please also read our response to comment 18 in which we provide a detailed explanation about scoping reviews), we were interested in all reported outcomes of paediatric outpatient treatment. A sentence added to the methodology to make this clear.
9	Did the 33 studies that were not 'a good fit' all fulfil the exclusion criteria in table 1? There are some studies in the Bryant & Katz systematic review that seem as though they should be included in this review from the inclusion criteria, but this may not be the case – it is just not completely clear from the methods.	All excluded studies did not meet the inclusion criteria and/or fulfilled the exclusion criteria. A greater level of detail has been reported in Table 1.
10	Because the methods are not clear on these points, it is therefore unclear whether the methodology is appropriate to answer the aim, especially the aim about key deficits in knowledge. ie any deficits found may have been published in other studies that were not included in this review's search.	Extra detail has been added to the methodology section.
11	There appears to be at least one missing study that should have been included by Orme et al Outpatient versus inpatient IV antibiotic management for pediatric oncology patients with low risk febrile neutropenia: a randomised trial. Pediatr Blood Cancer . 2014 Aug;61(8):1427-33. The primary and clear	We agree this paper should have been included and had been in our initial list of included papers. We have addressed this oversight.

	outcome was parent and patient experience with the sample size specifically calculated for that, and this may modify some of the discussion in this review.	
12	The authors state that 'Clinical complications such as line failures, rehospitalisation and adverse drug reactions were not reported as adverse events although these have been identified as such in the outcomes data extraction table', which appears to suggest that OPAT services are not reporting adverse events. This is inaccurate eg in Hodgson et al 'Table 5 Adverse events' specifically reports line complications, antibiotic complications and unplanned readmissions to hospital. It is not clear whether this is an oversight from one included study, whether this has occurred for multiple studies, or whether the authors mean something different.	We have added in the word 'consistently' to modify the global claim.
13	Readmissions to hospital are among the most critical outcomes when assessing an OPAT service, but the paragraph on this seems incomplete for several reasons: - the authors only categorize unresponsive to treatment, exacerbation of underlying condition and poor evolution of infectious disease, but there are other reasons for readmission reported in several studies eg line complications, antibiotic complications, an unrelated reason – why are these not included? - could the authors explain the difference between unresponsive to treatment and poor evolution of infectious disease - the above categories are only reported for 3 studies but these are reported in other studies too – why are the other studies not included?	The terminology used by the authors of the studies we reviewed was inconsistent and this made comparison problematic. We reported the subjective terms used by authors. We have added more detail and included the other studies into this section. Where specific percentages have been reported by the authors we have included these.
14	The paragraph on cost analysis is also flawed for a couple of reasons: - 'None of the studies undertook a comprehensive health economic evaluation' – since this is not the aim of a clinical description of an OPAT service, while true, this is an unwarranted criticism. A good health economic analysis would be an entirely different study. It's akin to saying none of the studies assessed variation in care using a health services delivery approach – true, but not the aim of the studies or were systematic in their approach' – could the authors clarify what this means and by whose definition? They may have been systematic but did not include all related costs eg costs to families - 'although these conclusions should be considered with caution' – this may be true but needs an explanation why? - 'However, cost effectiveness calculations did not account for the costs associated with complications (26) or the direct cost to families (6).' Is this true of all the studies that attempted to evaluate cost or just the two cited?	We agree with this comment and have edited text. (Note: within a scoping review the focus is on identifying gaps in the literature and areas that are under-represented (see also response to comment 18). However, we acknowledge your comments that criticising a study/studies for not achieving something that they had not set out to do can seem to be unfair. This was not our intention, our critique is of the body of literature as a whole We have therefore revised wording throughout to try and make this clearer). We have revised the text to clarify meaning.
15	'There is a lack of clarity and consensus as to what constitutes success in pOPAT, although this has not deterred authors from concluding that pOPAT appears to be safe (6,8,10,23,24).'	We have revised this statement to reduce the overly critical feel. It was not our intention to appear condescending.

	This sentence is again unnecessarily critical and comes across as condescending. If a service has low adverse events, low OPAT-related complication rates, low readmission rates and high rates of completion of treatment then why would the authors of this review not respect the authors of the original study's conclusion that OPAT is safe, even if their definition of success is different from another service or study?	(see also response to comment 18)
16	The paragraph on key knowledge deficits is true, but again, if these areas were not aimed to be addressed by the included studies, this is an unwarranted criticism: - 'We also know little about parents and children's experience of pOPAT. There is little reflection about the factors which may influence experience such as the child's age, nature of infection, family circumstances and the educational level of parents.' – this wasn't the aim of many if the included studies; in a study discussing clinical outcomes for cardiac patients, the parent and child experience would not be a requirement of a study unless it was a stated aim.	We agree that it was not the intention of most of the studies to include parents or children's perspectives. However, this remains a major deficit in knowledge that we identified from the scoping review. We have retained the section you quote in the paper but we have added a modifier into the opening sentence of the paragraph. (see also response to comment 18)
17	Additionally, considering the fact that infection has a higher incidence in families of lower socio-economic status (28,29), there is little detail about whether these families are excluded from pOPAT or, if in receipt, how they fare in comparison to families in better circumstances. – again, while this is important to know, this wasn't the aim of the included studies; in a study on the efficacy of an antibiotic to treat a particular infection, a socio-economic analysis would not be required if this wasn't the aim of the study, despite infections having a higher incidence in lower socio-economic groups.	Again as stated above we agree this was not the aim of the studies, what we are trying to present with this (as with parents/children point above) is that this is a gap that needs to be considered, particularly since knowing about socio-economics could usefully inform clinicians' decisions. (see also response to comment 18)
18	Is this called a scoping review because it isn't a PRISMA-compliant systematic review? (although the authors have stated systematic a couple of times)	The study is called a scoping review as we have used scoping review methodology (as per Arksey & O'Malley, 2005). This is a legitimate approach to synthesising research evidence about a broad topic including a broad range of study types. In brief it aims to map existing literature in terms of volume, nature and characteristics of primary research and to identify research gaps within the literature. It is commonly undertaken to determine value of undertaking a full systematic review and/or to determine the focus of a systematic review. A robust scoping review is systematic (i.e., rigorous etc.) but does not follow the same pathway as a systematic review as the intended outcomes are different

		(robust descriptive synthesis of evidence rather than focused synthesis of evidence with risk of bias addressed and with a primary focus on RCTs). It is possible that the reviewers lack of familiarity with scoping reviews (a relatively new approach) may explain the concerns raised about why our critique of papers raises issues that the studies had not intended to address.
19	pOPAT is an unnecessary and non-universal abbreviation. Once the authors have identified that the study is in children, the term OPAT can be used.	The abbreviation has been amended.
20	Suggest having the condensed summary chart as a supplementary table rather than 'available on request'.	The condensed summary chart will be included as supplementary data.
21	Suggest adding the type of quantitative study (eg record review, cohort, etc) to table 3.	The table presented as recommended by Pluye et al.
22	When listing the numbers of studies that reported particular factors eg home environment criteria, it would be more useful to the reader of the references to the specific articles could be included with statements so the reader can review these themselves. Without this, the reader is unable to relate the study with the specific factors addressed. Studies are cited in some sections but not others – it is inconsistent.	The results section has been amended as suggested and checked for consistency.
23	the authors would have been unable to determine whether some factors may not have been identified in a study because this was either not included by the authors of the study or editor of the journal it was published in as too obvious to include (eg parental compliance or suitable home environment) or not considered to need inclusion for other	We agree that we only accessed the material in the published paper (this is acceptable practice for a scoping review). However, parental compliance, suitability of home environment may either be seen as 'things that are too obvious to include' or maybe things that have been under-researched. The body of literature we scoped is heavily oriented to outcomes and targets that are clinician- and/or organisationally-oriented. The literature that does consider parents perspectives reflects a more nuanced perspective. However, within the word constraints of this paper we have not addressed this in detail.
24	the authors did not address comparison with inpatient care – although this was not stated as an aim, it should be mentioned in this paragraph as it is a potential limitation of any OPAT study without it reasons, rather than that the service not providing it.	We have added this in as a limitation

25	Little detail was provided about the outpatient settings or the actual suitability, difficulties or challenges of the home as a setting for pOPAT' If this was not the aim of the included study this is an unwarranted criticism.	As stated above, this is a comment on the body of literature. We have not changed this as this we believe that this statement is a statement of fact rather than unwarranted criticism. It feels fairly fundamental that some description of the environment would have been helpful.
26	'Adverse drug reactions (ADRs) were reported inconsistently and the definition of what constituted an ADR was not clearly defined.' This is not accurate. In some studies what constituted an ADR was very clearly defined, but may have been different between studies or not clearly defined in others. Suggest change to 'The definitions and reporting of ADRs was inconsistent between studies.'	We have changed this as per your suggestion.
27	'We also know little of children who were not selected for pOPAT or parents and children who declined this treatment and the reasons why.' This is in the paragraph on study quality but is unrelated to quality: a study can have high quality but if this is not one of the aims of the study then this is an unwarranted criticism.	We have moved this sentence to later in the discussion where it has a better fit.
28	'it should not be assumed that all families will choose pOPAT or that it will be the most appropriate intervention.'Did any of the studies state this, or is the authors' own assumption?	This is not our assumption; we are pointing out that without robust evidence from parents and/or children, we should not assume that this is what parents want. This is evident in van der Laag et al.'s study. We do not think this sentence is inappropriate within the context of the paragraph in which it appears and within the context of this being a scoping study.
29	The conclusion 'Future studies need to clearly define and use success indicators and adverse events in order to provide evidence that pOPAT is safe and effective' is unnecessary as the almost identical sentence appears three paragraphs above.	Conclusion has been reworded.
30	Neutropenia not neutropaenia.	This has been amended.
31	There is no reference for the RCT by Gupta et al.	This has been amended.
32	Inconsistent use of whole numbers or one decimal place for percentages.	This has been amended.